# Metabolic Factors Affecting Tumor Immunogenicity: What Is Happening at the Cellular Level?

**DOI:** 10.3390/ijms22042142

**Published:** 2021-02-21

**Authors:** Rola El Sayed, Yolla Haibe, Ghid Amhaz, Youssef Bouferraa, Ali Shamseddine

**Affiliations:** 1Global Health Institute, American University of Beirut, Beirut 11-0236, Lebanon; rola.m.sayed@gmail.com; 2Division of Hematology/Oncology, Department of Internal Medicine, American University of Beirut Medical Center, Beirut 11-0236, Lebanon; yh44@aub.edu.lb (Y.H.); ga116@aub.edu.lb (G.A.); yb21@aub.edu.lb (Y.B.)

**Keywords:** immunotherapy, checkpoint inhibitors, tumor microenvironment, immune-metabolism, glycolysis, OXPHOS, metabolic modulation, adaptation

## Abstract

Immunotherapy has changed the treatment paradigm in multiple solid and hematologic malignancies. However, response remains limited in a significant number of cases, with tumors developing innate or acquired resistance to checkpoint inhibition. Certain “hot” or “immune-sensitive” tumors become “cold” or “immune-resistant”, with resultant tumor growth and disease progression. Multiple factors are at play both at the cellular and host levels. The tumor microenvironment (TME) contributes the most to immune-resistance, with nutrient deficiency, hypoxia, acidity and different secreted inflammatory markers, all contributing to modulation of immune-metabolism and reprogramming of immune cells towards pro- or anti-inflammatory phenotypes. Both the tumor and surrounding immune cells require high amounts of glucose, amino acids and fatty acids to fulfill their energy demands. Thus, both compete over one pool of nutrients that falls short on needs, obliging cells to resort to alternative adaptive metabolic mechanisms that take part in shaping their inflammatory phenotypes. Aerobic or anaerobic glycolysis, oxidative phosphorylation, tryptophan catabolism, glutaminolysis, fatty acid synthesis or fatty acid oxidation, etc. are all mechanisms that contribute to immune modulation. Different pathways are triggered leading to genetic and epigenetic modulation with consequent reprogramming of immune cells such as T-cells (effector, memory or regulatory), tumor-associated macrophages (TAMs) (M1 or M2), natural killers (NK) cells (active or senescent), and dendritic cells (DC) (effector or tolerogenic), etc. Even host factors such as inflammatory conditions, obesity, caloric deficit, gender, infections, microbiota and smoking status, may be as well contributory to immune modulation, anti-tumor immunity and response to immune checkpoint inhibition. Given the complex and delicate metabolic networks within the tumor microenvironment controlling immune response, targeting key metabolic modulators may represent a valid therapeutic option to be combined with checkpoint inhibitors in an attempt to regain immune function.

## 1. Introduction

The introduction of immunotherapy to the treatment algorithms of malignant diseases has revolutionized the field of oncology with attention being shifted from off-target bombardment of tumor cells by using standard chemotherapy, to focused immune enhancement against tumor cells using vaccines, cytokines, adoptive cell therapy (ACT) and checkpoint inhibition. The concept of checkpoint inhibition lies within mounting and stimulating one’s immunity against cancer, through inhibiting discovered inhibitory checkpoints such as cytotoxic T-lymphocyte-associated protein 4 (CTLA-4), programmed cell death protein-1 (PD-1), and programmed cell-death protein ligand-1 (PDL-1) [1]. These inhibitory checkpoints impede the immune response and induce tolerance, thereby regulating the immunity to avoid excessive activation against one’s self through auto-reactivity. Nevertheless, these inhibitory checkpoints are harnessed by tumor cells as a means for immune evasion [2]. 

Multiple checkpoint inhibitors (CPI) have been implemented in practice, with remarkable responses that altered the treatment paradigm and improved disease prognosis in a number of hematologic and solid malignancies [3,4]. CPI have become the cornerstone of treatment of some diseases such as renal cell carcinoma (RCC), malignant melanoma, and non-small cell lung cancer (NSCLC) [5,6,7,8]. Unfortunately, the response to CPI is heterogeneous, and the success rate to therapy remains negligent in many circumstances [9]. Occasionally, tumors labeled as “hot”, with significant effector immune-cell activity, primarily attain significant tumor regression post CPI, but are soon rendered “cold” or “immune-resistant”, with recurrent tumor growth and disease progression. Whether patients have primary innate resistance (never-responders), or secondary acquired resistance, multiple factors are at play. 

Immune responses rely on intricate and dynamic interactions between malignant cells, immune cells and the surrounding tumor TME [10]. The TME represents the network of cells and structures surrounding tumor cells, including the extracellular matrix (ECM), vascularization, immune cells, and signaling molecules such as cytokines, growth factors and hormones [11]. Inflammatory and metabolic stimuli derived from the TME have a particularly important role in shaping and modulating the specific and innate immune responses on all levels of immune cells, thereby affecting tumor growth, metastasis, as well as response to treatment (Figure 1) [11,12]. 

Furthermore, the dysregulation of energy metabolism is a key player in the modification of the metabolic state and functional phenotypes of innate and adaptive immune cells infiltrating the TME [13]. Limited nutrient availability (glucose, amino acids, fatty acids, and oxygen) due to increased tumor cell consumption during active proliferation, obliges variable tumor-infiltrating immune cells that are competing over the same pool of nutrients, to adapt and shift to alternative metabolic pathways with distinct patterns of glucose and lipid metabolisms, consequently affecting cancer-related inflammation, as well as the pro-and anti-tumor immune-cell functions [13,14]. Aerobic and anaerobic glycolysis, oxidative phosphorylation (OXPHOS), and fatty acid biosynthesis (FAS) or oxidation (FAO) are all mechanisms of adaptation (Table 1). 

These metabolic adaptive mechanisms, along with involved inflammatory mediators, have a major influence on ICI resistance at the cellular level via drastic alteration of immune-cell crosstalk, leading to impairment of effector T-cell activation, and stimulation of regulatory immune cells such as regulatory T-cells (T-regs), TAMs, myeloid-derived suppressor cells (MDSCs), and tolerogenic DCs, etc.(Figure 2) [13,15].

In this review, we provide insight towards the inter-dependent immune-metabolic drivers of immunosuppression and resistance to immunotherapy, specifically checkpoint inhibition, both at the cellular level, within the TME, and at the host level, causing “hot” or “immunotherapy-sensitive” tumors to be “cold” or “immunotherapy-resistant” tumors.

## 2. Nutrients Affecting the Cellular Activity of Immune Cells in the Tumor Microbiome

### 2.1. Glucose Metabolism

During proliferation and tumor growth, cancer cells require a high demand for all nutrients, resulting in the depletion of sufficient nutrients needed for other tumor interstitial cells and immune cells within the TME [13,16]. Otto Heinrich Warburg was the first to tackle the idea of cancer cell aerobic glycolysis, or the “Warburg effect”, which is a mechanism of adaptation that provides tumor cells with the required energy needs, but results in elevated lactate and acidity of the TME [17,18] (Figure 3).

Upon T-cell activation, cellular energy needs increase, and T-cells exit the quiescent catabolic phase, and enter the effector phase by shifting to anabolic metabolism. The mTOR pathway is activated through the PI3K/Akt pathway, and T-cells resort to aerobic glycolysis, which further promotes IFN-γ secretion, and enhances the function of CD8+ T-cells [17]. Effector T-cells are noted to express high levels of glucose transporter GLUT1, in order to increase the glucose uptake needed in aerobic glycolysis. Furthermore, upon T-cell activation, sarco/ER Ca2+-ATPase (SERCA) activity is inhibited, and phosphoenolpyruvate carboxykinase 1 (PCK1) becomes overexpressed leading to excessive phosphoenolpyruvate (PEP) production. PEP, in turn, stimulates Ca2+-NFAT signaling and enhances T-cell effector functions. Acyl glycerol kinase (AGK) is another enzyme protein that promotes glycolysis and antitumor activity of CD8+ T-cells by inactivating PTEN and enhancing mTOR activity [19].

In brief, inhibition of mTOR would lead to suppression of glycolysis and FAO activation, resulting in impaired effector T-cell differentiation and enhanced memory phenotype [20,21]. Similarly, in situations of glucose starvation, aerobic glycolysis is limited. The Akt pathway is inhibited, IFN-γ gene expression is impaired, and pro-apoptotic B-cell lymphoma-2 (Bcl-2) pathways are activated endorsing T-cell apoptosis [22]. Moreover, insufficient glucose within the TME induces immunosuppression through decreased expression of zeste methyltransferase enhancer homolog 2 (EZH2), leading to impaired T-cell antitumor activity, decreased perforin, granzymes B and C, decreased cytokine production (mainly IFN-γ), and decreased T-cell viability [23,24,25,26]. 

The presence of PD-1 and CTLA-4 receptors has been noted to affect the glycolytic pathway through a decrease in glucose uptake, leading to impaired T-cell activation. However, PD-1 engagement alone showed to promote FAO and enhance lipolysis. These findings were supported in a preclinical mouse model, where PD-1 blockade reversed glucose restriction, allowing glucose influx and glycolysis with stimulated mTOR signaling, and eventual IFN-γ production, resulting in improved effector anti-tumor function. 

As for TAMS, they are divided into pro-inflammatory M1 macrophages and anti-inflammatory M2 macrophages. M1 macrophages are characterized with enhanced glycolysis, and attenuated TCAs, whereas M2 macrophages have complete TCA with enhanced FAO [27]. Differentiation is usually mediated by tumor-derived lactate via Akt/mTOR signaling pathway affecting the expression of pro-inflammatory cytokines and chemokines, Arg1, IL4Ra, and epithelial-to-mesenchymal transition (EMT) [28,29]. 

Inhibition of glycolysis with 2-deoxyglucose may disrupt metastatic tendency and M2 phenotype behavior; however, mTOR inhibition of glycolysis tends to cause abnormal vascularization promoting metastasis [30]. 

When it comes to NK cells, activated NK cells rapidly produce IFN-γ to exert effector functions, requiring significant energy supplies and depending on both glycolysis, as well as oxidative phosphorylation [31]. Glucose is of crucial importance to NK cell activation, and NK cells express 3 types of glucose transporters GLUT1, 3 and 4, of which GLUT1’s expression is significantly increased upon activation, promoting glucose uptake [32]. Moreover, sterol regulatory element binding proteins (SREBPs) are regulators of glycolysis in NK cells [33]. FBP1 is another important effector of glycolysis in NK cells. It is a key enzyme in the gluconeogenesis pathway, highly expressed in tumors, and whose inhibition restores glycolysis, as well as function of NK cells, ultimately inhibiting tumor progression [34]. In addition, glucose metabolism can also be affected by the presence of excess TGF-β in the TME, leading to the inhibition of mTOR signaling, and eventual NK-cell suppression [35].

Activated DCs have high levels of glucose metabolism. In mice, bone-marrow-DCs (BMDCs) rapidly induce glycolysis through PI3k/AKT/mTOR/HIF-1α signaling cascade after exposure to lipopolysaccharides, thereby increasing the rate of glycolysis and lactic acid production [36]. In the presence of glucose-deficient media or the glycolysis inhibitor 2-deoxyglucose, the activation of BMDCs including the expression of CD80, CD86, and CCR7, and the secretion of pro-inflammatory cytokines are significantly impaired [37,38].

### 2.2. Amino-Acid Metabolism

In addition to glucose metabolism, amino acid metabolism is indispensable for immune-cell activation and differentiation. Tumor cells and immune cells compete over amino acids such as tryptophan (trp), glutamine and L-arginine [39].

Trp catabolism is involved in physiological immune suppression through the tryptophan–kynurenine–aryl hydrocarbon receptor (Trp–Kyn–AhR) pathway, and is involved in acquired and intrinsic resistance to immunotherapy [40,41]. Activated T-cells are extremely sensitive to the concentration of Trp in the peripheral environment, which triggers the effector T-cell apoptosis [42]. Moreover, kynurenine, a metabolite of Trp, acts as a ligand that activates the aryl-hydrocarbon receptor, and suppresses CD8+ T-cells, as well as NK cell activity [43]. TAMs up-regulate the indoleamine 2,3-dioxygenase (IDO) involved in Trp metabolism, leading to Trp depletion, and kynurenine accumulation, with resultant inhibition of tumor immune response. Furthermore, IFN-γ-stimulated DCs also have an increased IDO-1 expression and activity. Thus, effector T-cells activated by DCs might suppress DCs’ function as a negative feedback [13,44,45]. 

Glutamine, another essential amino acid, plays a critical role in effector T-cell generation through glutaminolysis. Glutaminolysis, on the other hand, allows ATP production to support T-cell development and functionality by increasing IL-2 receptor expression, and cytokine production via ERK/MAPK-coordinated regulation of glutamine uptake [46,47]. Competition for glutamine, in conjunction with HIF1a stabilization, affects the synthesis of S-2-hydroxyglutarate, causing an overall change in histone methylation and promoting CD8+ T-cell differentiation and effector function [48]. Furthermore, in TAMs, the expression of glutamine transporter is in favor of M2 polarization [49].

L-arginine promotes human T-cell proliferation, but limits its differentiation, maintaining a more T-cell memory phenotype through Interferon-alpha/beta receptor alpha chain (IFNAR1), which inhibits the transcription of metabolic genes such as Ornithine transcarbamylase (OTC) and Argininosuccinate Synthase 1 (ASS1). It also induces the production of transcription factors such as Bromo-domain-adjacent-to-zinc-finger-domain 1B (BAZ1B), PC4-and-SFRS1-interacting-protein 1 (PSIP1), and Translin (TSN). These T- memory cells rely less on glycolysis and promote OXPHOS [50,51]. 

In TAMs, arginine metabolism is the most studied. L-arginine metabolism in macrophages leads to nitric oxide (NO) production using the inducible NO synthase (iNOS), which eventually leads to the suppression of enzymes needed in TCA-ETC pathways, and promotes glycolysis as well as M1 phenotype transformation. This same NO production plays a role in the suppression of cytotoxic NK cell activity. In NK cells, glutamine and L-arginine affect mTOR signaling, and consequently regulate c-Myc expression, which is necessary in IL-2/IL-12 production, essential for NK functionality [52,53]. In conditions of hypoxia and increased acidity of the TME, Arg1 is over-expressed, with consequent increased conversion of L-arginine into L-ornithine, further involved in proline and polyamine synthesis, a characteristic of M2 phenotype polarization [54]. Moreover, multiple Th2 cytokines such as IL-4, IL-13, and TGF-β can induce Arg-1 expression in myeloid cells, including macrophages and DCs [55].

### 2.3. Lipid and Fatty Acid Metabolism

Lipid metabolism appears to be increasingly relevant in modulating cancer-related inflammation, expansion of myeloid cells and reprogramming of inflammatory phenotypes [56].

In the TME, with respect to TAMs, the M1/M2 polarization states are dependent on different patterns of lipid metabolism (FAS vs. FAO respectively) [57]. TAMs are known to change their lipid metabolism according to adaptation needs. For example, lipopolysaccharide-stimulated macrophages have M1 phenotype, and show enhanced synthesis of fatty acids (FA), and triglycerides, with induction of pro-inflammatory cytokines and increased phagocytosis. Macrophages activated by IL-4 are usually M2 macrophages [58,59]. They have stimulated transcriptional activator 6 and peroxisome proliferator-activated receptor gamma coactivator-1 beta (PGC-1β), and exhibit increased triglyceride uptake through CD36 with enhanced FAO [60]. In addition, TAMs highly express epidermal fatty acid binding protein (EFABP), capable of promoting lipid droplet formation as well as IFN-β production, responsible for enhancing the recruitment of tumoricidal effector cells [61]. On the other hand, TAMs can also be responsible for immune tolerance by promoting CCL2 and IL10 production through eicosanoids and 15-lipoxygenase-2 [62,63]. Another mechanism of TAM reprogramming is the triglyceride hydrolysis pathway, where the abhydrolase-domain-containing 5 (ABHD5) in TAMs inhibits the accumulation of reactive oxygen species (ROS), which in turn reduces C/EBPε-dependent spermidine production, and ultimately promotes tumor growth [64,65].

Activated T-cells use aerobic glycolysis, whereas memory cells resort back to oxidative phosphorylation, requiring fatty acid oxidation to produce ATP [66]. Moreover, cholesterol metabolism is highly associated with T-cell activity, and the modulation of cholesterol metabolism by targeting the cholesterol esterification enzyme ACAT1 can potentiate the antitumor response of CD8+ T-cells [67]. Furthermore, Th17 and Tregs are dependent on lipid metabolism for their survival and function, with fatty acid-binding protein-5 shown responsible for the activation of the IFN signal in Treg cells, and responsible for IL-10 modulation [68].

NK cell FA and cholesterol synthesis through FA synthase and stearoyl-CoA desaturase-1 (FASN/SCD1), and 3-hydroxy-3-methylglutaryl-CoA synthase/acetyl-CoA acetyl-transferase 2 (Hmgcs1/Acat2) respectively, result in FA accumulation, which leads to immune signal activation [69]. SREBPs transcription factors regulate this immune signal activation by IL-2, as well as IL-12 present within the TME [70,71]. Furthermore, adiponectin, which is secreted by adipose tissue, is involved in the metabolism of glucose and FA in various types of cells, including NK cells, and is responsible for the regulation of their maturation and activation [72].

Finally, de novo lipid biosynthesis is an important metabolic process after BMDCs activation, whose maturation is dependent on the production of citrate, acetyl CoA and lipids [73]. However, Wnt signaling can integrate PPARγ-regulated FAO, and drive DC into tolerization, with eventual immune evasion [74]. Also, lipid accumulation in tumor-associated DCs, mediated by upregulation of scavenger receptor A (SR-A1 or MSR1), can negatively regulate antigen presentation via MHC class II [56,75].

## 3. Mechanisms and Pathways Modulating Metabolism and Affecting Cellular Activity of Immune Cells in TME

### 3.1. Hypoxia, HIF-1 α and ROS

Cellular metabolism is dependent on oxygen supply provided by tumor vasculature. Uncontrolled tumor cell proliferation depletes oxygen and leads to hypoxia. As an adaptation mechanism, hypoxia-inducible factor 1α (HIF-1α) is produced, aiding in adaptation to low tissue oxygen levels, in addition to the development of more aggressive, treatment-resistant cancers (Figure 4) [76,77]. 

Hypoxia promotes immune effector cell apoptosis, reduces the generation and release of inflammatory cytokines, and supports the generation of immunosuppressive cells such as M2 macrophages as well as Tregs, through C-C Motif Chemokine Receptor 6 (CCR6) stimulation, via C-C Motif Chemokine Ligand 20 (CCL20) [79,80].

Another important modulator of tumor immune-metabolism is the production and metabolism of the highly unstable reactive oxygen species (ROS). Formed mainly in mitochondria, ROS are involved in the regulation of multiple biological processes, including the regulation of phosphatases, kinases, transcription factors, epigenetic regulators and antioxidant enzymes [81]. Interestingly, ROS have been involved in T-cell subset differentiation. Moderate ROS production in T-cells is required for cellular, and signaling processes leading to T-cell activation, whereas high levels of ROS in the environment have shown a tendency towards Th2 phenotype shifting. Accumulation of mitochondrial ROS eventually leads to exhausted T-cell state [82,83]. However, low levels of ROS have been associated with Th1 and Th17 cell differentiation [84,85]. Within the TME, which is usually characterized by elevated ROS levels, the use of antioxidants has been shown to impair T-cell responses [86].

ICI through PD-1 blockade increases cellular ROS assisting in the proliferation and activation of CD8+ T-cells in the tumor microenvironment [87]. Pharmacologic interventions to increase ROS may effectively synergize with PD-1 blockade to enhance cytotoxic effector T-cell activity. 

### 3.2. Pyruvate Kinase Muscle Splicing

A key feature of cancer cells that distinguishes them from normal cells and accounts for their rapid proliferation is their metabolic regulation, mainly through shifting towards the distinct aerobic glycolysis, namely the Warburg effect [88]. Pyruvate kinase (PK), a glycolytic enzyme that catalyzes the conversion of phosphoenolpyruvate (PEP) and ADP to pyruvate and ATP, has been implemented in this metabolic switch [89]. PK has four isoforms distributed in different tissues according to the cell type. Specifically, the PK isoforms M1 (PKM1) and M2 (PKM2) are produced by alternative splicing from the same single PKM gene [90]. PKM1 is expressed in tissues requiring a large amount of ATP, such as brain and muscle, whereas PKM2 is mainly expressed in proliferating cells, such as embryonic cells and tumor cells [91].

Structurally, PKM1 forms a stable tetramer with high affinity for PEP, and subsequently produces pyruvate to be used in OXPHOS. Conversely, PKM2 exists both as a tetramer with high affinity for PEP, yet requiring allosteric regulation, and as a dimer with low affinity for PEP, and thus low catalytic efficiency [90]. The characteristic reduced metabolic activity of the dimeric PKM2 probably accounts for its high expression in proliferating cells, causing a decrease in PEP clearance and a subsequent increase in glycolytic intermediates for synthetic processes [92].

The tumor metabolic phenotype is regulated by the expression of either PKM1 or PKM2. In fact, switching from PKM2 to PKM1 greatly reduces lactate production in tumor cells, and is associated with a markedly reduced tumor growth [89]. The two isoforms are generated through mutually exclusive alternative splicing of the PKM pre-mRNA, reflecting inclusion of either exon 9 (PKM1) or exon 10 (PKM2) [93]. Hence, the regulation of PKM alternative splicing is crucial for understanding tumor metabolic regulation.

For instance, heterogeneous nuclear ribonucleoproteins (hnRNP), which represent RNA binding proteins with well-established roles as sequence-repressors of splicing, were found to be directly involved in the alternative splicing event [94]. Particularly, three hnRNP proteins, polypyrimidine tract binding protein (PTB, also known as hnRNPI), hnRNPA1 and hnRNPA2, bind repressively to sequences flanking exon 9, resulting in exon 10 inclusion. The transcription of these proteins is upregulated by the oncogenic transcription factor c-Myc, leading to a high expression of PKM2 as compared to PKM1, and thus presenting a selective advantage for tumor formation [93].

### 3.3. Adenosine Pathway (A2A Stimulation)

The production of adenosine lies in two significant pathways. The first pathway generates adenosine via the hydrolysis of extracellular ATP (eATP) by ectonucleotidases such as CD39 and CD73 [41]. The second pathway produces eAMP by a reaction that uses NAD+ as a substrate, catalyzed by the activity of both of CD38 (an NAD+ ectohydrolase) and CD203a (an ectonucleotide pyrophosphatase). Subsequently, CD73 catalyzes the hydrolysis of eAMP into adenosine [95]. 

Adenosine binds to four P1 G-protein coupled receptors: A1, A2A, A2B and A3 [41]. It has distinct effects on immune cells through its A2A receptor. It impairs the activation, proliferation, survival and cytokine production of T lymphocytes. The activation of A2A in T-cells stimulates the differentiation of CD4+ T-cells into Tregs that co-express both CD39 and CD73 and are capable of generating further adenosine and stimulating A2A and A2B, thus contributing to the suppressive functions of Tregs [96]. 

In B cells, A2A activation causes the suppression of NF-kB, resulting in impaired signaling downstream of the B cell receptor, activation and survival of B cells [97]. In macrophages, the expression of A2A and A2B is upregulated through Toll-like receptor (TLR) signaling. Once A2A is activated, macrophages are drawn to differentiate into a pro-tumor and tolerogenic phenotype. This phenotype is associated with increased immunosuppressive cytokine production such as Il-10 and VEGF [98]. Likewise, in dendritic cells, adenosine signaling aids in cytokine expression of IL-10, TGFβ, and VEGF, while suppressing the production of pro-inflammatory cytokine IL-12, and co-stimulatory molecules such as CD80 and CD86. In neutrophils, A2A activation inhibits migration and production of ROS [95]. In mast cells, A2A or A2B activation prevents mast-cell dependent vasoconstriction, and limits chemotaxis and degranulation [97]. 

The effect of adenosine on tumor cells involves not only its immunosuppressive impact on immune cells, but also its stimulation of tumor growth and metastasis. The latter is manifested by A2A activation that mediates PI3K-AKT signaling or A2B that mediates ERK, JNK and/or p38 MAPK signaling in tumor cells. Consequently, it has been shown in preclinical models of breast, gastric, liver, bladder or renal cancers that these mechanisms promote tumor cell survival, proliferation, adhesiveness and invasiveness [97]. 

Tumor stroma is another crucial element involved in cancer progression. It was shown that CD39 and CD73 are expressed in abundance by cancer-associated fibroblasts (CAFs), making these stromal cells capable of generating immunosuppressive adenosine in several types of cancers such as breast, ovarian, colorectal and pancreatic origin. CAFs are able to contribute to the pro-tumorigenic effects of adenosine by activating the adenosinergic-signaling pathway in the cells themselves. In cases of high-grade serous ovarian cancer, a correlation was made between high levels of CD73 expressed on CAFs and poor prognosis, as CD73 expression impairs antitumor immunity. Furthermore, poor prognosis was associated with high abundance of CAFs in colorectal cancer (CRC). In a different study, CAFs were seen to undergo proliferation upon A2A receptor activation in NSCLC. In addition to CAFs, CD73 and adenosine receptors are expressed in abundance on vascular endothelial cells [99]. They both regulate vascular permeability, neo-angiogenesis, and lymphocyte trafficking. Findings of preclinical studies suggest that T-cell trafficking to tumor cells is suppressed by down-regulating adhesion molecule expression, due to adenosine production by endothelial CD73. It has also been shown that CD73 promotes angiogenesis via adenosine signaling in tumors by promoting the VEGF secretion. Moreover, the formation of new lymph vessels and tumor draining lymph nodes was seen in mouse models as a result of the A2A signaling pathway [100]. 

In brief, activating adenosine signaling pathway in TME destabilizes and disrupts antitumor immunity [101]. This antitumor immunity is manifested through: adenosine being generated by CD39/CD73, providing suppression signaling through A2A and/or A2B in NK cells and CD8+ cells, and inhibiting the production of effector immune molecules in lymphocytes such as granzymes and perforin, IFNγ and TNF, while inducing the production of tolerogenic factors such as TGFβ and IL-10 in myeloid cells [97,102,103]. Moreover, adenosine signaling through A2A allows Treg proliferation, the expression of immune checkpoint receptors such as PD-1, CTLA4, and the secretion of immunosuppressive factors such as IL-10 and TGFβ. Lastly, in preclinical models, the expression of CD39 and CD73 on Tregs stimulated tumor growth [96,97]. 

### 3.4. Lactic Acidosis

Several products from the metabolism of tumor cells accumulate in the TME and affect immune cell function. The highly glycolytic tumors have high lactate levels in their microenvironment that may reach up to 40 mM, produced by the lactate dehydrogenase A (LDHA) [104,105]. The monocarboxylate transporters co-transport lactate and protons outside the cell; hence, it decreases the PH and leads to the acidification of the extracellular environment, with deleterious effects on immune cells [106,107]. This acidification of the TME will lead to a decrease in the monocyte differentiation, prevention of NK cell activation, loss of immune surveillance and tumor growth. It affects innate immune cells, and decreases INF production in tumor infiltrating T-cells [108,109,110]. 

A low environmental pH may not affect cell viability, but it will lead to a decrease in the function of cytotoxic effector cells and a decrease in the production of cytokines as first described by Fischer et al. [111,112]. A pH of 6.6 leads to an alteration in cell-cycle progression, a decrease in cytokine secretion, and an altered expression of IL-2 receptors leading to diminished T-cell activation and proliferation [113]. In melanoma patients, it was noted that a pH of 6.5 causes a decrease in the expression of TCR component, a decrease in the secretion of IFN, TNF and IL-2, and a decrease in tumor infiltrating T-cell responsiveness [114].

High levels of lactate can stabilize HIF1 and induce lactylation of histones, leading to the polarization of anti-inflammatory M2 macrophages [115,116]. It can also decrease CD1 expression, maintain a tolerogenic phenotype and impair migratory response to chemokine derived from lymph nodes in DCs. 

A study conducted by Calcinotto et al. showed that the administration of a high dose of proton pump inhibitor to melanoma-bearing mice leads to a decrease in the TME pH with resultant increase in T-cell infiltration and reversion of anergy [114]. Pilon-Thomas et al. reported that the use of bicarbonate to neutralize tumor acidity increases T-cell infiltration in TME [117]. Renner et al. showed that the restriction of acidosis and glycolysis in melanoma mouse models preserves T-cell effector function and enhances the effect of CPI therapy [118].

### 3.5. Concept of Extracellular Vesicles

Extracellular vesicles (EV) comprise a diverse group of membrane vesicles secreted by different cell types that may affect the metabolism and function of immune cells [119]. Tumors, for example, are able to secrete EVs that can have biological effects on immune cells [120,121]. TAMs, as another example, play a role in anti-tumor immune phenotype by releasing EVs that contain enzymes involved in lipid metabolism [122].

Pancreatic cancer cells secrete EVs containing microRNAs in hypoxic conditions that cause progression of cancer and favor poor prognosis by activating PI3K signaling pathway and inducing M2 macrophage polarization [123].

PD-L1 can be found on the surface of EVs produced by tumor cells and plays a role in suppressing the immune response [121,124]. Exosomal PDL-1, a form of EV PDL-1 plays a role in predicting the response with CPIs [125]. Poggio et al. reported that blocking exosomal PDL-1 will lead to activation of T-cells in the lymph nodes, therefore suppressing tumor cell proliferation [126].

EV does impact immunotherapy response but the effect on the function and metabolism in anti-tumor immunity is still uncertain.

## 4. Other Mechanisms of Adaptation of Immune Response

### 4.1. Role of Sphingosine Kinase-1

The sphingo-lipid metabolism is an important element of cancer cell metabolism. Different enzymes are involved in the formation of cancer metabolites including sphingosine-1-phosphate (S1P) [127]. Sphingosine kinases (SK) are responsible for the development of S1P by phosphorylation of sphingosine into S1P [127]. Multiple human cancers harbor the SK type 1 isoform that is coded by the gene SPHK1, and is responsible for the high levels of S1P. SK1/S1P pathway is a key element in the control of cancer cell proliferation, as well as apoptosis, invasion and angiogenesis [127]. The expression of SPHK1 was mainly studied in melanoma. Its level of expression seemed highest in metastatic states, and therefore, the progressive state of melanoma was directly correlated to the levels of SPHK1 gene, where models with higher levels of SPHK1 expression revealed shorter progression free survival when treated with anti-PD1 therapy [127]. Using yumm1.7 cells that mainly blocked SK1 resulted in enhancing the response to anti-PD1 treatment in melanoma patients. SK1 silencing leads to an increased ratio of CD8+ to Treg cells, and consequently increases the cytokines and interleukins supporting immunosuppression, and decreasing Foxp3+CD4+T-cells [127]. Thus, the down regulation of SK1 could enhance the response to ICI (anti-CTLA4 and anti-PD1 therapy) (Figure 5) [127]. 

S1P, as a lipid second messenger, works by signaling through the G protein-coupled receptors (S1P_1–5_) [128]. This signaling pathway is correlated with multiple physiological activities including: vascular growth, homeostasis of the central nervous system, and the biology of the lymphocytes, specifically cell trafficking [128]. S1PR modulation has been discovered to be responsible for the activity of FTY720, a chemical reform of the fungal derivative Myriocin, extensively studied for its role in provoking lymphopenia and prolonged allograft survival in various species [128]. Furthermore, using fetal liver from S1PR1 deficient embryos to create bone marrow chimeric mice, Matloubian et al. proved that the outlet of lymphocytes from the thymus and secondary lymphoid organs did not take place in the absence of S1P signaling. This emphasized the importance of S1P–S1P_1_ interaction in regulating lymphocyte development [129]. Low levels of S1P, or inhibition of S1P1 causes depletion of lymphocytes. Inversely, S1P1 infusion showed an increase in lymphocyte production [128]. The ability to maintain lymphocytes in lymph nodes, or permit their allocation to the blood, is an important feature in launching an effective immune response. 

Upon studying the interaction of S1P1 with CD69, a surface activating marker causing retention of lymphocytes that is upregulated in response to interferons, a reciprocal regulation system was noted, with each one being mutually expressed on the cell surface [129]. Furthermore, T-cell activation is modulated through a balance between C-C chemokine receptor type 7 (CCR7), a chemokine receptor on T-cell cortex, and S1P1 boosting signal [129]. The chemokines C-C motif ligand (CCL19) and CCL21 are attracted by CCR7. CCL19 causes desensitization to CCR7 signaling, while high exposure to S1P leads to S1P1 internalization, rendering cells unresponsive to circulating signals in the blood or lymph [128]. Losing CCR7 signaling can increase the time T-Lymphocytes stay inside lymph nodes [128]. In absence of S1P1, individuals lose the ability to inhibit the proliferation, and activity of T-regs [130]. On the contrary, overexpression of S1P1 decreases the quantity and activity of T-regs, with consequent decrease in immunosuppression. In T helper type 17 (Th17) cells, S1P1 signaling, which is amplified through a Janus-activated kinase 2 (Jak2)-dependent manner, activates STAT3 resulting in Th17 stimulation [128,130]. On the other hand, T resident memory (Trm) cells positive for CD69, have a low expression of KLF2 and its target, S1P1r [128,130]. Thus, the expression of KLF2 in CD8+ T-cells ends up with an increase in S1P1, and consequently, a decrease in CD69, proving the reciprocal role of S1P1 and CD69 on Trm [128]. Through the interaction of the different receptors and chemokines with S1P1, it is shown how T lymphocyte trafficking, migration and differentiation is linked to S1P1 where S1P1 modification may alter immunogenicity, and affect tumor prognosis as well as response to therapy [128].

### 4.2. Role of MUC-1 Mucin

Multiple other proteins have also been involved in the immune system, including a group of high molecular-weight glycoproteins called mucin. Mucin have been assembled into 2 groups: the trans-membrane deposits comprising of MUC1, MUC3, MUC4, MUC10-18, and the soluble (gel forming) glycoproteins including MUC2, MUC5AC, MUC5B, MUC6, MUC7, MUC8, MUC9 and MUC19 [131]. One of these mucins, the polymorphic epithelial mucin (PEM) or MUC1, is found to be expressed in all human epithelial cells of adenocarcinomas as well as multiple hematological malignancies [131,132]. During translation, this protein undergoes cleavage. It has a tandem repeat (TR) with an external domain indirectly linked to the membrane of the MUC-1-C short extracellular domain (EC) as well as to the transmembrane and cytoplasmic domains [131]. Through the TR and the known excess of serine and threonine residues, MUC-1 is quite potent for O-glycosylation [131,132]. It is therefore crucial to differentiate the mucin protein on normal tissue in comparison to neoplastic one. The loss of the well-organized structure of the gland rendering it without demarcation of the apical and basolateral epithelial cells yet with ultimate expression of MUC1 is characteristic of adenocarcinoma [132]. The MUC1 will be then differentiated through the process of glycosylation and over expression on tumor cells signifying the importance of this protein in the maintenance of the tumor and its progression [131,132]. Consequently, the number and the type of MUC1 post glycosylation on tumor cells plays a role in stimulating the immune response where its over-expression can inhibit cell-lysis mediated by cytotoxic lymphocytes, giving tumor cells the ability to escape the immune system [132]. It has been noted that in multiple cancers, although antibodies against MUC1 increase in the blood, and act in a cellular mediated response, cancer cells are not eliminated. Suggested reasons would be the inefficiency of the immune response at different points: poor antigen presentation, toxic TME, abnormal T-cell activation response [132]. Many efforts have been made to surpass this faulty immune response through different mechanisms of conjugation and co-stimulation; yet, no ultimate solution has been achieved. However, MUC1 is still considered a possible targetable option to enhance the fight against tumor cells. 

### 4.3. Effect of Acetyl-CoA Carboxylase 1(ACC1), and Mitochondrial Involvement

Digging deeper in the cell organization at the level of cellular metabolism, it is noted that naïve T-cells require a low rate of metabolic activity supplied by the mitochondrial metabolism, to suffice the energy requirements for their survival. The ATP required is mainly generated through the TCA cycle by the process of oxidation of pyruvate and fatty acids (FAO) as well as OXPHOS [133]. Through the TCA cycle, glucose is used to get enough energy supply where activated T-cells save other amino acids and fatty acids for further growth and expansion [133]. On the contrary, Th17 cells depend highly on acetyl-CoA carboxylase 1 (ACC1) which is the main enzyme in de novo fatty acid synthesis [133]. Targeting ACC1 was suggested as a pathway to modulate the immune system especially in autoimmune diseases and inflammatory conditions [133]. Discussing what is important in the metabolism of memory T-cells, it was established that mitochondria have an extracellular capacity in energy generation under stress named: spare respiratory capacity (SRC) [133]. This is crucial for memory CD8+ T-cells differentiation. Having more mitochondria and boosted SRC permits T-cells to have a rapid response to different triggers. Basic helix-loop-helix-family member e40 (Bhlhe40) is a newly discovered essential element needed for mitochondrial fitness that is capable of improving mitochondrial capabilities, enhancing the immune system functionality, and adding to the role of immunotherapy [133]. 

## 5. Effect of Environmental Factors at the Host Level

Multiple environmental and host factors such as the presence of inflammation with cytokine release, obesity and metabolic syndrome, caloric deficit, gender, infection, microbiota and smoking can play a significant role in modulating the anti-tumor response of immune checkpoint inhibitors (Figure 6).

### 5.1. Cytokine Release: Inflammation and Autoimmune Response

Cytokines in the TME play an essential role in the recruitment and activation of immune cells. While some cytokines potentiate and support a strong immune response, others have suppressive effects favoring tumor resistance and progression [134]. The immunosuppressive cytokines could be secreted by the tumor itself or the surrounding macrophages in order to suppress the anti-tumor response [135]. By stimulating Tregs and inhibiting cytotoxic T-cells (CTL), TGF-β plays an essential role in this immunosuppression, thus conferring resistance to CPI [136]. Once stimulated, Tregs in turn can suppress the anti-tumor immune response by inhibiting effector T-cells through direct contact or secretion of IL10, IL35 or even TGF-β [137]. Moreover, IFN-γ is another cytokine involved in the response and resistance of tumors to CPI response and resistance. In fact, IFN-γ was found to increase the response to PD-1 and CTLA-4 blockade by up-regulating tumor antigen presentation on MHC-I and PDL-1 expression.

VEGF is another cytokine that can also have immunosuppressive properties leading to resistance to CPI. In fact, VEGF levels were found to be higher in patients who fail to respond to CPI, and the addition of VEGF-inhibitors was found to reverse resistance to immunotherapy [138,139]. This could be explained by the ability of VEGF to promote extravasation of Tregs into the TME while decreasing the infiltration of CTLs [140]. In addition, MDSCs were also found to promote resistance to CPI, and the inhibition of their trafficking into the TME was found to enhance the response of rhabdomyosarcoma to CPI in mouse models [141]. Finally, TAMs were also involved in enhancing the resistance to CPI as their inhibition restored the response to T-cell checkpoint immunotherapy in pancreatic cancer models [142].

### 5.2. Obesity and Metabolic Syndrome

Controversial results exist about the role of obesity in cancer and CPI resistance. Obese patients were found to have a chronic cytokine-driven low-grade inflammation leading to T-cell exhaustion with a decrease in NK and effector T-cells but an increase in PDL-1 expression [143]. In addition, the elevated leptin levels in obese patients were found to recruit MDSCs in the TME favoring immunosuppression [144].

While it is usually associated with decreased survival in cancer patients, the “obesity paradox”, a phenomenon positively correlating elevated BMI and survival, plays a role in the response of tumors to CPI in particular [145,146]. In a study done by McQuade et al., obesity was associated with higher progression-free and overall survival in metastatic melanoma patients, especially in men treated with targeted or immunotherapy [147]. This finding was highlighted in several other studies where a positive association was reported between elevated BMI and survival in melanoma patients treated with either anti-PD1, anti-CTLA4 or a combination of both [136,147,148]. This could be at least in part attributed to the inability of a fatty liver to effectively clear the therapeutic antibodies increasing their bioavailability, in addition to the chronic low-grade inflammation and the increased PDL-1 expression reported in people with elevated BMI [143]. 

### 5.3. Caloric Deficit

Caloric deficit was found to play an important role in the function of the immune system. Intermittent fasting for example induced a higher response in cancer patients treated with CPIs [149]. This is explained by the ability of intermittent fasting to increase the anti-tumor activity through activation of CTLs and reprogramming of TAMS [149]. Caloric restriction was also found to improve the signaling between immune cells and adipose tissue in mouse models [150]. In addition, by decreasing insulin like growth factor 1, IL6, anti-OX40 immunotherapy and m-tor signaling, caloric deficit allowed for a decrease in T-cell senescence thus improving immunosurveillance [151,152,153,154]. 

### 5.4. Gender Effect

In a study done by Ye et al., differences in some immune checkpoint protein expression, including PDL-1, were noted between males and females [155]. This difference could explain the discrepancy in response to specific CPI between genders. In fact, Wu et al. showed in a meta-analysis that the effect of ICB on overall survival and progression free survival was higher in males than in females, especially in the case of CTLA-4 blockade [156]. Further studies need to be done to further assess the role of gender in the response and resistance to CPI.

### 5.5. Infection

CPI has not been widely studied in human infectious diseases. As immunosuppressive therapies lead to T-cell exhaustion, many pathogens make use of those pathways in order to evade the immune system. For example, malaria pathogenesis was shown to depend on the PD-1 pathway, and targeting PDL-1 improved protection against the disease [157]. In addition, immunotherapy was found to play a role in the treatment of chronic viral infections like chronic HIV or Hepatitis B. In those cases, continuous antigen exposure leads to exhaustion of T-cells that will overexpress CTLA-4 and PD-1/PDL-1. As such, targeting immune checkpoints could potentially help in the cure of such chronic infectious conditions [157]. The use of CPI in the treatment of COVID-19 has also been controversial. While the use of CPI in the first pre-infectious phase can make patients more resistant to the virus, the use of those immunostimulant antibodies may exacerbate the cytokine storm during the infection, thereby increasing the severity of the disease [158]. Finally, the use of CPI has been reported to cause several infectious adverse events, especially in cancer patients. These include pneumonias, intra-abdominal infections, as well as varicella, pneumocystis pneumonia, invasive aspergillosis and even tuberculosis [159].

### 5.6. Microbiota

The intestinal microbiota has been reported to play an essential role in immunosurveillance, and some studies have shown that the microbial flora within the gut can interact with the tumor antigens to stimulate the pattern recognition receptors to produce different cytokines with various effects on the immune system function [160]. This immunomodulatory effect allowed the gut microbiota to influence the response of patients to CPI. Frankel et al. showed that patients with melanoma had better responses to immunotherapy when Bacteroides caccae was present in their gut flora [161]. Further supporting this point, Chalabi et al. showed that administration of PPI or antibiotics 30 to 60 days prior to the treatment reduced the response of NSCLC patients to treatment with Atezolizumab [162]. In fact, antibiotics could preferentially kill the sensitive microbial species while PPI may reduce the gastric pH, both leading to significant changes in the composition of the microbes within the gut [163,164]. Therefore, recent studies are using microbial ecosystem therapeutics or fecal transplantation in combination with CPI in the treatment of melanoma, with the latter showing improvement in the response to immunotherapy [165].

### 5.7. Smoking

Different studies have shown controversial results regarding the role of smoking in the response to CPI. In a study published in JAMA Oncology, Lee et al. concluded that smoking status did not influence the sensitivity of NSCLC patients to treatment with CPI [166]. On the other hand, different studies have shown that in NSCLC, smokers had a benefit tendency to anti-PD1 therapy as opposed to non-smokers [167,168,169]. In fact, smoking is thought to increase the mutational load within the tumor, increasing the oncogenic neoantigen expression, which leads to the activation of an effective anti-tumor immune response [170]. In addition, Kerdidani et al. found that DCs exposed to emphysematous TME have increased PD-L1 expression, leading to immune tolerance and escape, with a potentially stronger response with anti-PDL1 treatment [170]. Whatever the relationship is between smoking and the response to CPIs, this habit should not be appraised. Non-smokers still have longer overall survival as compared to smokers [171].

## 6. Metabolic Manipulation at the Genetic and Pharmacologic Levels

Metabolism is an important element that could play a crucial role in cancer immunotherapy, introducing a new domain in the field of immune metabolism. Having control over the metabolism of cells can boost one’s immunity against tumor cells, working in line with available CPI [172]. It is important to realize the connection between tumor cell and immune cell compartments living in the TME, as well as the different nutrient sensing mechanisms and the different metabolic switches. All these factors play a significant role in controlling the responses of the immune system to tumors with or without CPI [172]. 

First of all, it is important to consider the phenotypic criteria of different immune cells, and their cytotoxic requirements, as well as the specific metabolic expertise needed in enhancing the immune response and decreasing immunosuppressive behavior [172]. The secretion of different cytokines like IL-2, TNFα, IFN γ will be lost progressively as inflammatory effector immune cells are progressively exhausted, and eventually go through apoptosis [172]. The apoptotic process includes a reduction in the immune-system functionality that occurs through PD-1 expression, known as the “exhaustion marker” [172]. To overcome immune system exhaustion, anti-PDL-1 (atezolizumab, durvalumab, and avelumab) or anti-PD-1 (pembrolizumab and nivolumab) are used, but it is not enough. Tumors lose immunogenicity, and become cold due to the aforementioned adaptive metabolic mechanisms. Targeting these metabolisms may provide a key to finding means of reverting the immune process. Nevertheless, although some metabolic alterations may be grouped into pro- or anti-tumor favoring sides, however, the reality is far more complex. One metabolic pathway may be immunogenic in a cell group, and rather tolerogenic in a different cell group, with the involvement of a multiplex of chemical intermediaries.

At the genetic and pharmacologic levels of anti-tumor immune response, many manipulations can occur and many pathways and therapeutic approaches are being studied. One example is Itraconazole (ITZ), a 35-year-old antifungal, that works through the inhibition of lanosterol 14-α-demethylase (14LDM), to decrease ergosterol production in fungi, and cholesterol in mammals [173]. It has been investigated in multiple studies for its role as an anti-cancer medication in different cancer types: basal cell carcinoma, prostate cancer, gastric cancer, and NSCLC [173]. It acts through the suppression of inflammation, and by inducing cell-cycle arrest through apoptosis and autophagy, as well as by suppressing angiogenesis and bypassing drug resistance [173]. TME is characterized by an excess of inflammatory cytokines, such as: tumor necrosis factor-α (TNF-α), interleukin-1β (IL-1β), a variety of signaling pathways, such as: transforming growth factor-beta (TGF-β), and platelet derived growth factor (PDGF) [174]. TNF-α and IL-1β cause an increase in the transcription factor glioma-associated oncogene homolog (GLI) present in tumor cells, and consequently activate the hedgehog (Hh) pathway [174]. It was shown that ITZ works as well through the inhibition P-glycoprotein (P-gp) resulting in a decrease in the inflammatory cytokine secretions [174].

Hh is another pathway investigated in anti-tumor immune response. It is non-active most of the time [174]. There is a seven-pass transmembrane protein called smoothened receptor (SMO) that is inhibited by a twelve-pass transmembrane protein, the Patched1 (PTCH1) [174]. Following the inhibitory signaling pathway, GLI converts to a complex with kinesin protein (Kif7) and suppressor of fused (SUFU) [174]. Then, protein kinase A (PKA) and glycogen synthase kinase-3 (GSK3) phosphorylate this complex, causing degradation through ubiquitination. Eventually, GLI will block target gene transcription [174]. 

Hh ligands are released upon exposure to an external factor, or a change in the environment similar to cancer states, and then bind to PTCH1, leading to the activation of the Hh pathway. Recovery of SMO regulates the release of GLI from the cytoplasmic inhibitory protein, permitting GLI to enter the nucleus and consequently bind DNA and initiate target gene transcription [174]. Target genes include: *Cyclin-D1*, and *MYC,* known for their role in proliferation, *BCL-2,* known for its importance in apoptosis, *ANG1/2* and its significance in angiogenesis, *SNAIL* and its worth in EMT, and *NANOG*, *SOX2* with their prominence in stem cell self-renewal. 

J. Kim et al. elaborated that ITZ works on SMO and consequently inhibits the Hh pathway [174]. Through the inhibition of SMO and GLI1, ITZ causes the repression of the phosphorylation of class III phosphatidylinositol 3-kinase (PI3K), and AKT, a serine/threonine protein kinase which is part of the PI3K/AKT/mTOR pathway, that is major in the tumorigenesis [174]. ITZ inhibits angiogenesis by suppressing vascular endothelial growth factor (VEGF) signaling, through glycosylation of VEGF receptor 2 downstream Hh pathway and GLI1 [174].

Moreover, ITZ has shown to play an important role in drug resistance, which is an important element of cancer management. Drug resistance was based on the ATP-binding cassette (ABC) transporters, in conditions with high expression of EMT and highly activated Hh pathway [174]. ITZ was able to bypass drug resistance by suppressing the ABC through the inhibition of Hh pathway [174].

Metformin, an old anti-diabetic agent, has also proven its role in anti-tumor immunity, causing the extension of CD8+memory T-cells. It works by inhibiting m-TOR signaling downstream from the AMPK pathway, and by inducing the transformation from a glucose-dependent anabolic state (effector T-cell) to a catabolic state of metabolism (memory T-cell) [172]. This way, T-cells are boosted by quality (functionality) and quantity [172]. Metformin may represent an important solution to overcome the apoptotic process of CD8+ TILs, and the decrease in cytokine production [172]. It acts as a constraint to PDL-1 expression through AMPK activation, leading to PDL-1 phosphorylation, thus prompting its degradation through the endoplasmic reticulum in cancer cells, and consequently enhancing the activity of cytotoxic T-cells [172].

In addition, metformin may play a role in neutralizing other immune-inhibitory cell groups within the TME including TAMs, Tregs and MDSCs [172]. The TME that is lactate-rich and glucose-deficient, works on damaging T-cell function, and activating TAM M2 anti-inflammatory phenotype, consequently inducing tumor angiogenesis and metastasis with immune tolerance [172]. Metformin has proven efficient in inhibiting the transformation of TAMs into their M2 phenotype, ultimately suppressing tumor invasion [172]. It proved to work as well on diminishing neutrophils and polymorphonuclear MDSCs (PMN-MDSCs), reprogramming them into OXPHOS and thus consequently leading to tumor growth inhibition [172]. Metformin also works by reducing the transcription factor FOXP3 responsible for Treg cells differentiation, and inhibiting their re-generation [172]. 

Another important player in tumor immunogenicity, gut microbiota with its different microorganisms, has been shown to influence the immune system. Examples include: Akkermansia muciniphila that enhances CD8+ T-cell activity, and boosts anti-PD1 treatment effect, and Faecali-bacteria, as well as Bifido-bacteria that govern the anti-inflammatory effect, inhibiting overt activity of the immune system [172]. Giving metformin to mice has shown to cause a significant change in the gut microbiota, inducing increased bacterial existence, with resultant increased levels of CTLA-4 presentation, rendering them more sensitive to CTLA-4 blockers [172]. Many trials regarding the possibility of combining metformin with immunotherapy are ongoing. 

Multiple other means of affecting metabolic pathways are possible, such as the previously mentioned attempts of high-dose PPI, bicarbonate infusions, high-dose ascorbic acid administration, targeting IDO function in protein metabolism, ACAT 1 in lipid metabolism, or ROS for synergy with PD1 blockade [175,176,177,178,179]. However, these approaches are not well-established and need to be studied further.

## 7. Conclusions

Tumorigenesis is highly dependent on the surrounding TME where multiple immune metabolic factors play a significant role in the crosstalk, modulation and reprogramming of infiltrating immune cells. Both at the cellular and host levels, different conditions sway immune cells towards pro- or anti-inflammatory phenotypes. The TME represents a dynamic, and intricate network of a multiplex of adaptive mechanisms, allowing the co-existence of tumor and immune cells, competing over limited nutrients, in varying biochemical storms of mediators.

At the cellular level, nutrient depletion, hypoxic conditions, TME acidity and different inflammatory mediators, such as: cytokines, interleukins, growth factors or hormones; all take part in determining and defining the nature of immune response. Glucose, amino acid, and lipid metabolism can be very different in one cell population or the other, signifying notable cellular plasticity and contributing to tumor aggressiveness and CPI resistance. Glycolysis or OXPHOS, FAS or FAO, IDO suppression or overexpression, Arg1 suppression or overexpression, etc. all can have a wide range of effects on tumor immunogenicity. The difficult part is to be able to better understand these vast mechanisms as a whole in the wider picture, and how they interact amongst each other, to be better able to act upon different factors, and improve patient survival with better tumor responses.

At the host level, chronic inflammatory conditions, obesity, caloric deficit, smoking, infections and microbiome status, can all be involved in modulating the immune response, with life-style modifications and prevention still playing an important role in cancer management.

Finally, targeting key modulators of immune metabolism and immune activation may represent valuable possibilities of therapeutic interventions to couple with ICI and improve response rates, rendering initially “hot-tumors” that have become “cold” hopefully “hot” once again. Drugs like ITZ and metformin may have already proven their role in modifying tumor immunogenicity, but multiple other targeted “precision-medicine” approaches are showing great promise of benefit and efficacy. Nevertheless, to better treat, we need to better understand.

## Figures and Tables

**Figure 1 ijms-22-02142-f001:**
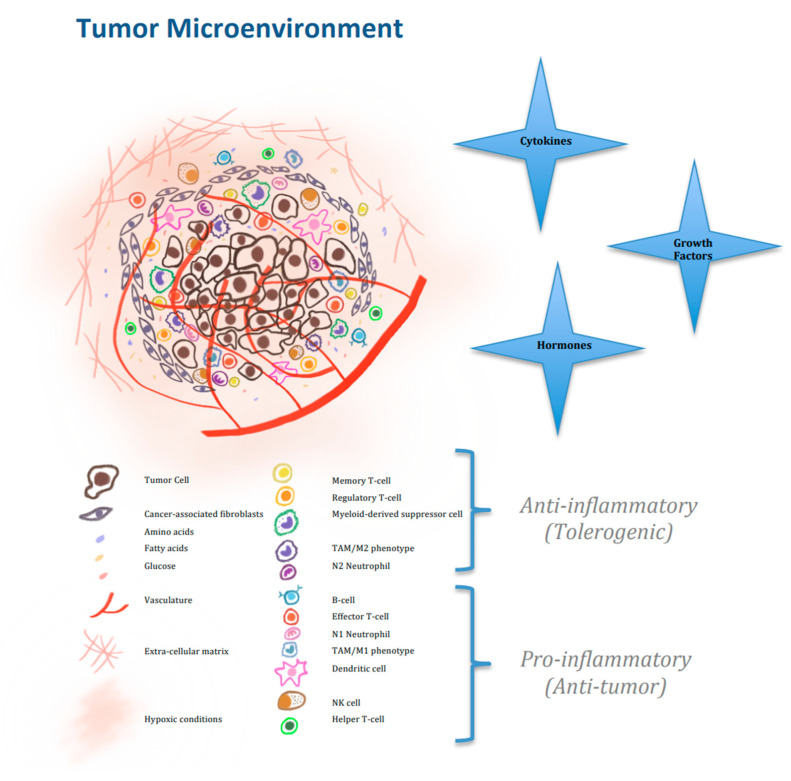
Multiple constituents of the tumor microenvironment.

**Figure 2 ijms-22-02142-f002:**
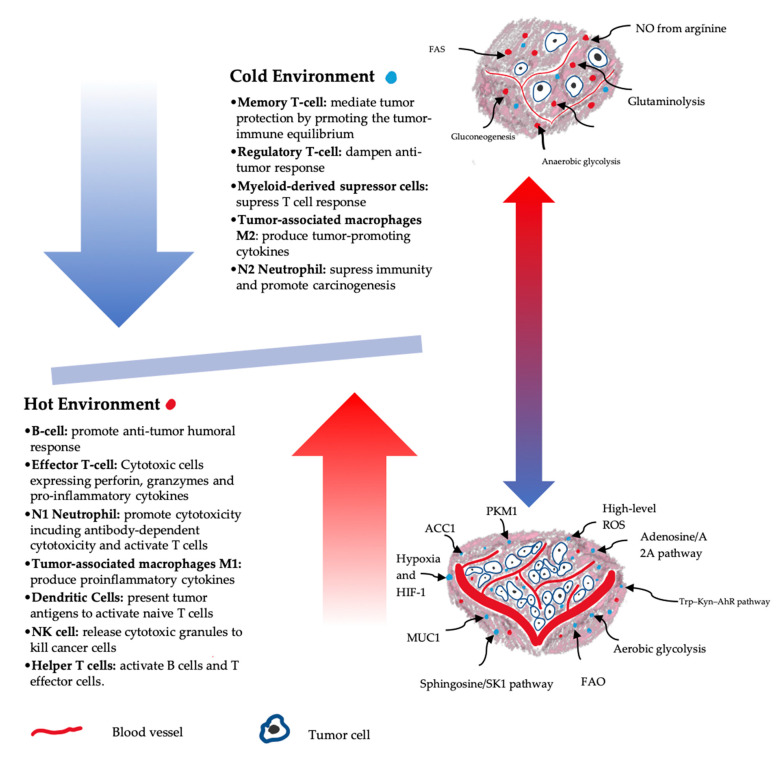
**Hot vs. Cold tumor microenvironment**. The profile of immune cells within the tumor microenvironment can switch the balance between a hot or “immune-sensitive” tumor and cold or “immune-resistant” tumor. Nutrient metabolisms and deficiencies, hypoxia, acidity, and different secreted inflammatory markers lead to modulation of immune-metabolism and reprogramming of immune cells towards pro- or anti-inflammatory phenotypes.

**Figure 3 ijms-22-02142-f003:**
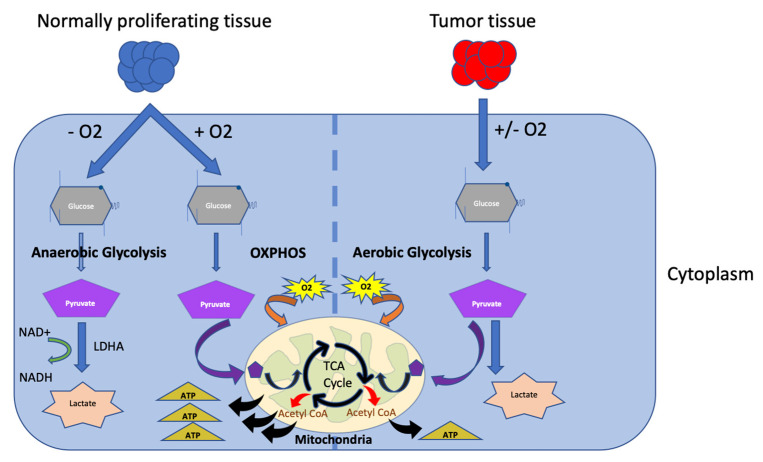
**Warburg Effect**. Normally, anaerobic glycolysis takes place in the cytoplasm, where glucose is converted to pyruvate, then consequently to lactate via the lactate dehydrogenase (LDHA), concomitantly producing nicotinamide adenine dinucleotide hydrogen (NADH). Within mitochondria, in presence of oxygen, pyruvate is converted to acetyl-CoA through Kreb’s cycle or TCA cycle. This process takes place in a series of reactions within an electron transport chain (ETC), producing adenosine triphosphate (ATP). It is known as oxidative phosphorylation or OXPHOS, and in presence of excess oxygen, it is called aerobic glycolysis or the “Warburg effect”.

**Figure 4 ijms-22-02142-f004:**
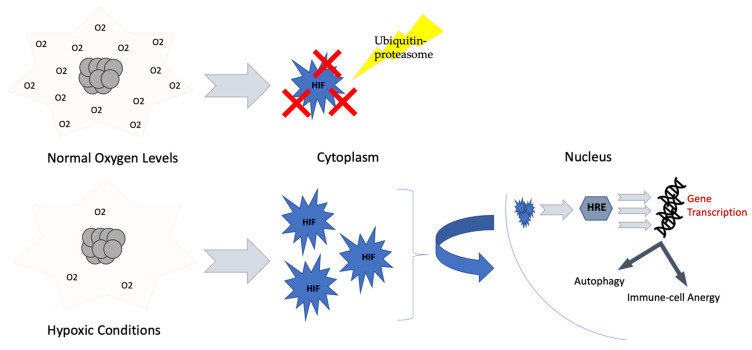
**Hypoxia and HIF-1*****α.*** In normal oxygen levels, HIF-1α is rapidly degraded by the ubiquitin-proteasome system. With hypoxia, HIF-1α accumulates in the cytoplasm, and is translocated to the nucleus, where it forms the hypoxia-responsive element (HRE) responsible for the transcription of genes involved in various cellular pathways, including autophagy as well as immune-cell anergy [78].

**Figure 5 ijms-22-02142-f005:**
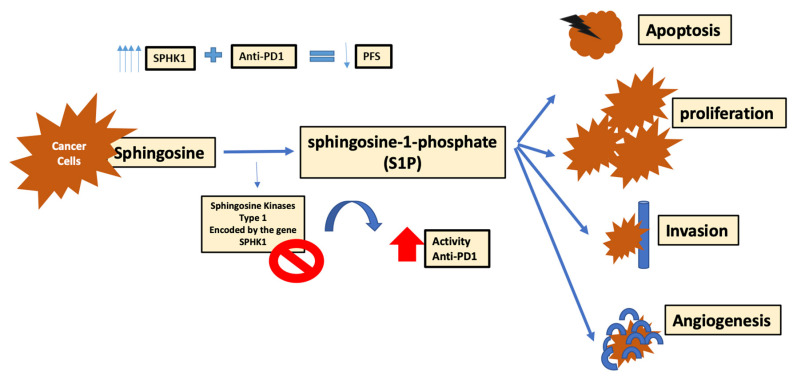
Sphingosine resistance mechanism PFS (progression free survival).

**Figure 6 ijms-22-02142-f006:**
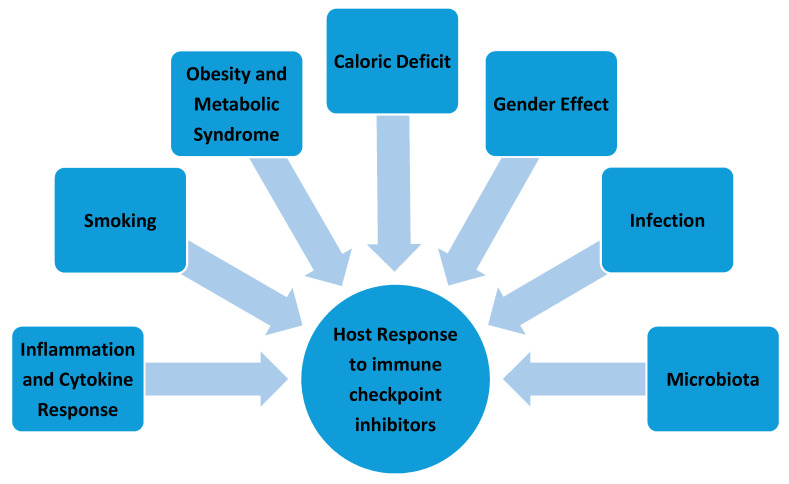
Environmental and host factors affecting the anti-tumor response of immune checkpoint inhibitors.

**Table 1 ijms-22-02142-t001:** Nutrients and metabolic pathways modulating the activity of immune cells and the antitumor immunity.

Moderators of Immune Cells’ Activity and Metabolism	
Nutrients	Other Mechanisms and Pathways
• **Glucose** o Increased glucose uptake through up-regulation of GLUT receptors o Aerobic and anaerobic glycolysis (Warburg effect) o Resultant acidotic TME with excess pyruvate o Pro-and anti-inflammatory phenotypes of immune cells dependent on glucose provision • **Amino Acids** o Required for activation and differentiation of immune cells o Role of Trp-Kyn-AhR pathway in intrinsic and acquired immunotherapy resistance o Trp metabolism, IDO and immunosuppression o Glutaminolysis, ATP production and effector T-cell Function/M2 TAM polarization o L-Arginine promotes proliferation and limits differentiation of effector T-cells through IFNAR1 • **Lipids** o Modulate cancer-induced inflammation, and reprogramming of inflammatory cytokines o LPS and Tg metabolism affect TAMs’ activity profile o Memory cells rely on FAO o Cholesterol metabolism is associated with T-cell activity o Fatty acid and cholesterol synthesis are involved in NK activity o Maturation of BMDCs relies on de novo lipid biosynthesis	• **Hypoxia, HIF-1 α and ROS** o Hypoxia promotes effector cell apoptosis, reduces cytokines and activates Tregs o Moderate ROS levels allow T-cell activation, signaling and differentiation o High ROS levels lead to T-cell exhaustion o Low ROS levels are associated with Th1 and Th17 differentiation • **Adenosine** o Adenosine impairs activation, proliferation, survival and cytokine production by T lymphocytes using A2A receptor o Adenosine favors Treg proliferation and expression of PD-1 and CTLA4 • **Lactate** o Acidification decreases monocyte differentiation, prevents NK cell activation and affects innate immunity by decreasing INF production o Acidification decreases the function and cytokine secretion of effector T-cells **Extracellular Vesicles** o Impact tumor response to immunotherapy but their role in antitumor immunity is uncertain • **Others** o Sphingosine Kinase-1 o MUC-1 Mucin o Acetyl-CoA Carboxylase ACC1

## Data Availability

Not applicable. No new data were created or analyzed in this study. Data sharing is not applicable to this article.

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
