# Peer review of "Metabolic Factors Affecting Tumor Immunogenicity: What Is Happening at the Cellular Level?"

_ijms, 2021, doi:10.3390/ijms22042142_

Round 1

Reviewer 1 Report

This manuscript covers a broad spectrum of mechanisms interacting with the immunue-checkpoint inhibits at the cellular level. Most of the descriptions of the mechanism are informative. The section 3.3 on adenosine pathway, however, would be better expressed if the whole section is focused and summarized a bit. 

Author Response

Dear Reviewer,

The authors would like to thank you for the time and effort placed into reviewing our manuscript entitled: Immune-checkpoint Inhibitors: hot to cold tumors again; what's happening at the cellular level?

We are very glad that you have liked our manuscript, and we are quite thankful for the high evaluations you have given us.

Regarding the comments and suggestions:

Comments:

This manuscript covers a broad spectrum of mechanisms interacting with the immunue-checkpoint inhibits at the cellular level. Most of the descriptions of the mechanism are informative. The section 3.3 on adenosine pathway, however, would be better expressed if the whole section is focused and summarized a bit. 

Reply:

Thank you for your comments. We tried our best to include all the relevant information regarding the metabolic mechanisms affecting tumor immunogenicity within the microenvironment, influencing the phenotypic behavior of immune cells, as well as the crosstalk between them, to shift from pro- to anti-inflammatory settings and vice versa, in interaction with checkpoint inhibition. We included few more figures for better graphical elaboration of some of the mechanisms (Figure 2 page 4, Figure 3 page 5, Figure 4 page 8).

Regarding section 3.3 discussing the adenosine pathway, you are absolutely correct. We have removed all the un-necessary details and attempted to summarize and focus this part on how the A2A pathway alters immune cell trafficking and disrupts anti-tumor immunity.

Best regards,

Ali Shamseddine,

Department of Internal Medicine, Division of Hematology-Oncology

American University of Beirut Medical Center, Beirut, Lebanon

PO Box 11-0236, Riad El Solh 110 72020

Tel +961 1374,Fax +961 1 370 814,

Reviewer 2 Report

This is a well written comprehensive review.

Given the review focuses on metabolism they might work metabolism into the title. A diagram indicating what aspects favour hot or cold tumors would also help.

They could mention the role of sphingosine kinase in T resident T cells. egSkon CN...Jameson SC  or Park CO,Kupper TS

No other comments

Author Response

Dear Reviewer,

The authors would like to thank you for the time and effort placed into reviewing our manuscript entitled: Immune-checkpoint Inhibitors: hot to cold tumors again; what's happening at the cellular level?

We appreciate your interest in our manuscript, and quite thankful for your evaluation.

Regarding the comments and suggestions:

Comments:

This is a well written comprehensive review.

Given the review focuses on metabolism they might work metabolism into the title. A diagram indicating what aspects favour hot or cold tumors would also help.

They could mention the role of sphingosine kinase in T resident T cells. egSkon CN...Jameson SC  or Park CO,Kupper TS

No other comments

Reply:

Thank you for your valuable input. We have tried to be all inclusive, and as comprehensive as possible in our review.

We have changed the title as per your recommendations to:

“Metabolic Factors Affecting Tumor Immunogenicity: what’s happening at the cellular level?”

Perhaps this way, the title would better portray the main focus of the manuscript: different inter-dependent metabolic mechanisms at play within the tumor microenvironment, swaying immune cells into pro-, or anti-inflammatory phenotypes, and affecting the “cold” or “hot” behavior of tumors.

A diagram was also added (Figure 2 page 4), named: “Hot vs Cold tumor microenvironment”. It portrays the profile of immune cells within the TME, switching between “hot” and “cold” phenotypes, with the different mechanisms involved leading to each.

Regarding the role of sphingosine kinase in resident T cells, we added to section 4.1, a part on the effect of sphingosine-1- phosphate (S1P1) on T cell trafficking, as well as the interaction of S1P1 with CD69, CCR7 and CCR19, and its effect on the quantity and activity of circulating T cells.

(paragraphs 2 and 3 of the section- pages 12 and 13: [ S1P, as a lipid second messenger....as well as response to therapy.])

Best regards,

Ali Shamseddine,

Department of Internal Medicine, Division of Hematology-Oncology

American University of Beirut Medical Center, Beirut, Lebanon

PO Box 11-0236, Riad El Solh 110 72020

Tel +961 1374,Fax +961 1 370 814,

Reviewer 3 Report

In their review „Immune-checkpoint Inhibitors: hot to cold tumors again; what’s happening at the cellular level?” Sayed and co-workers give a comprehensive review of metabolic factors influencing tumour immunogenicity. On the whole I liked the review and certainly found it informative; however, when I had finished reading it, I was left unsure what the target audience and the main message were. I think refocusing and sharpening would greatly benefit this work and make it more accessible.

  1. Title: The title actually pretty much summarizes the main weakness of this paper. It is long and bulky and does not really reflect the focus of the actual paper. Maybe the authors could consider something along the lines of: “Metabolic factors controlling tumour immunogenicity”?
  2. The authors should consider what their target audience is, I doubt anyone reading this paper will not have at least a working knowledge of the Warburg effect; yet the authors spend a paragraph explaining it. A paragraph is too long for people already knowledgeable and far too brief for an in-depth discussion. Also, in other sections the authors discuss in great details pathway information which bulks down the flow of their argument and leaves the reader without orientation, without adding much to the overall discussion. I would estimate that about a third of the paper’s length could be cut, in addition some pathway descriptions etc. might be better suited in graphical form.
  3. In contrast to what I just said, section 6 which is the part of therapeutically interest and section 7 should be expanded. Following the descriptive biological part, the practical (therapeutic) part needs to serve as a counterpart, while the conclusions should contain a critical assessment of the field.
  4. The English is good and comprehensible, but occasionally slightly stilted and a few times the wrong words are used and prepositions and articles are occasionally missing. A wee polish by a native speaker after re-writes would improve the reading experience.
  5.  

On the whole I’d like to reiterate that I enjoyed the paper and there is certainly a place for such a review in the literature, I’d just like to urge the authors to further improve the focus and sharpen their line of argument.

Author Response

Dear Reviewer,

The authors would like to thank you for the time and effort placed into reviewing our manuscript entitled: Immune-checkpoint Inhibitors: hot to cold tumors again; what's happening at the cellular level?

We appreciate your elaborate input and relevant comments, and we are quite thankful for your evaluation.

Regarding the comments and suggestions:

Comments:

In their review „Immune-checkpoint Inhibitors: hot to cold tumors again; what’s happening at the cellular level?” Sayed and co-workers give a comprehensive review of metabolic factors influencing tumour immunogenicity. On the whole I liked the review and certainly found it informative; however, when I had finished reading it, I was left unsure what the target audience and the main message were. I think refocusing and sharpening would greatly benefit this work and make it more accessible.

Reply: We appreciate your interest in our review, as we have attempted to be as inclusive and comprehensive as possible. We tried our best to include all the relevant information regarding the metabolic mechanisms and pathways affecting tumor immunogenicity within the microenvironment, and influencing the phenotypic behavior of immune cells, as well as the crosstalk between them, shifting those cells from pro- to anti-inflammatory settings and vice versa. The main target was to help better understand tumor immune metabolism, to be better able to target such pathways in the future, possibly enhancing tumor responses to immunotherapy within the context of “precision-medicine”. We attempted to target all readers who may have interest in the topic: scientists, biologists, immunologists, oncologists and physicians, etc. However, we understand your point, and fully agree with you, as occasionally the text has been un-necessarily extensive, leading to loss of the main focus and goal of the manuscript. We have reviewed all sections, deleted non-informative data, and rephrased some parts, in an attempt to make the manuscript more focused.

  1. Title: The title actually pretty much summarizes the main weakness of this paper. It is long and bulky and does not really reflect the focus of the actual paper. Maybe the authors could consider something along the lines of: “Metabolic factors controlling tumour immunogenicity”?

Reply: We have changed the title as per your recommendations to:

“Metabolic Factors Affecting Tumor Immunogenicity: what’s happening at the cellular level?”

Perhaps this way, the title would better portray the main focus of the manuscript: different inter-dependent metabolic mechanisms at play within the tumor microenvironment, swaying immune cells into pro-, or anti-inflammatory phenotypes, and affecting the “cold” or “hot” behavior of tumors.

  1. The authors should consider what their target audience is, I doubt anyone reading this paper will not have at least a working knowledge of the Warburg effect; yet the authors spend a paragraph explaining it. A paragraph is too long for people already knowledgeable and far too brief for an in-depth discussion. Also, in other sections the authors discuss in great details pathway information which bulks down the flow of their argument and leaves the reader without orientation, without adding much to the overall discussion. I would estimate that about a third of the paper’s length could be cut, in addition some pathway descriptions etc. might be better suited in graphical form.

Reply: We attempted to target all readers who may have interest in the topic: scientists, biologists, immunologists, oncologists and physicians, etc. Considering the wide range of various levels of targeted audience, we tried to be elaborate, yet not too extensive. However, we see your point, as too much elaboration and too many details would obscure the bigger picture and decrease the quality of the message to be delivered. As such, we have tried to delete some unnecessary information (example: section 3.3 pages 9 and 10 summarized), as well as rephrase other parts, some of which were represented in graphical forms as suggested (Figure 3 page 5 describing “Warburg Effect” with the elaboration removed from the text; Figure 4 page 8 describing “Hypoxia and HIF-1a”, with the description of mechanisms summarized in the text). Unfortunately, we could not cut down a third of the paper’s length.

  1. In contrast to what I just said, section 6 which is the part of therapeutically interest and section 7 should be expanded. Following the descriptive biological part, the practical (therapeutic) part needs to serve as a counterpart, while the conclusions should contain a critical assessment of the field.

Reply: Section 6 “Metabolic Manipulation at the Genetic and Pharmacologic Levels” pages 17 and 18, was rephrased, re-organized, and expanded, to clarify the main message of the manuscript, tackling the multiplex of metabolic mechanisms interacting at the cellular level to influence tumor immunogenicity. We attempted to help readers understand the possibility of targeting such pathways, at times with simple old drugs as itraconazole and metformin, others by adding unusual drugs such as proton pump inhibitors, high-dose ascorbic acid, and bicarbonate infusions, and finally, by using complex mechanisms to block different metabolic pathways such as targeting IDO function, ACAT1 and ROS. All of these approaches need to be further studied.

The conclusion was also changed and expanded to better explain the main message and make sure the reader will be left with a clear perspective of the complexity of the tumor environment, and the dynamic nature of the immune cells contributing to the immunogenicity of tumors, as well as the possibility of specifically targeting these immune metabolisms to enhance immune activation and tumor responses.

  1. The English is good and comprehensible, but occasionally slightly stilted and a few times the wrong words are used and prepositions and articles are occasionally missing. A wee polish by a native speaker after re-writes would improve the reading experience.

Reply: We tried to go through the entire text and adjust the language accordingly, making sure of grammatical correctness.

On the whole I’d like to reiterate that I enjoyed the paper and there is certainly a place for such a review in the literature, I’d just like to urge the authors to further improve the focus and sharpen their line of argument.

Reply: Thank you so much for your extensive comments and critical opinion. We are very glad you have enjoyed our paper. We believe your constructive criticism significantly contributes to the amelioration of our manuscript. We hope we have made all the proper adjustments needed.

Best regards,

Ali Shamseddine,

Department of Internal Medicine, Division of Hematology-Oncology

American University of Beirut Medical Center, Beirut, Lebanon

PO Box 11-0236, Riad El Solh 110 72020

Tel +961 1374,Fax +961 1 370 814,

Round 2

Reviewer 3 Report

Thank you for this extensive Revision, I find the (already good) review much improved and am looking Forward to reading your future work.